# RNA Interference Reveals the Impacts of *CYP6CY7* on Imidacloprid Resistance in *Aphis glycines*

**DOI:** 10.3390/insects15030188

**Published:** 2024-03-13

**Authors:** Shuangyu Li, Hongjia Yang, Yixiao Wang, Lisi Wei, Jiawei Lyu, Zhimeng Shan, Xinxin Zhang, Dong Fan

**Affiliations:** College of Plant Protection, Northeast Agricultural University, Harbin 150030, China; bilingualli@163.com (S.L.); 18800426621@163.com (H.Y.); 15663583528@163.com (Y.W.); wls253312@icloud.com (L.W.); lvjw0412@163.com (J.L.); 15545577009@163.com (Z.S.)

**Keywords:** *Aphis glycines*, CYP6, detoxification, RNA interference, insecticidal efficacy

## Abstract

**Simple Summary:**

The soybean aphid *Aphis glycines* is a globally present agricultural pest that is increasingly developing resistance to pesticides, causing huge economic losses. Cytochrome P450 monooxygenases (CYP450s) are major detoxifying enzymes that metabolize plant toxins and insecticides. In this study, we evaluated the relative expression levels of the CYP6 families in response to imidacloprid treatment using qRT-PCR. The gene *CYP6CY7* was significantly increased after treatment induction. Using a transdermal delivery system for nanocarriers to carry siRNAs and detergents that attach to the nota of aphids, this gene was successfully altered, thus significantly decreasing the gene’s P450 activity by up to 21.96%. A further bioassay showed that the mortality of *A. glycines* due to imidacloprid treatment increased by 14.71%. Overall, our findings indicate that *CYP6CY7* might detoxify imidacloprid in *A. glycines*, providing a theoretical basis for the further study of the mechanisms of action of CYP6s and potential new methods for improving insecticidal efficacy.

**Abstract:**

Cytochrome P450 (CYP) is a group of important detoxification enzymes found in insects related to their resistance to insecticides. To elucidate the CYP6 family genes of P450, which are potentially related to imidacloprid resistance in *Aphis glycines*, the CYP6 cDNA sequences of *A. glycines* were studied. The transcriptome of *A. glycines* was constructed, and the CYP6 cDNA sequences of *A. glycines* were screened. Their relative expression levels in response to imidacloprid induction were examined through qRT-PCR, and the CYP6s with higher expression levels were used to study the detoxification of imidacloprid through RNA interference and a bioassay. Twelve CYP6s were obtained from the *A. glycines* transcriptome. These samples were named by the International P450 Nomenclature Committee and registered in GenBank. After 3, 6, 12, 24 and 48 h of induction with LC50 concentrations of imidacloprid, the relative expression levels of these CYP6s increased; the expression level of *CYP6CY7* experienced the highest increase, being more than 3-fold higher than that of those of the non-imidacloprid-induced CYP6s. After RNA interference for *CYP6CY7*, the relative expression level of *CYP6CY7* significantly decreased after 3, 6 and 12 h, while the corresponding P450 enzyme activity decreased after 12 and 24 h. The mortality of *A. glycines* due to imidacloprid treatment increased by 14.71% at 24 h. *CYP6CY7* might detoxify imidacloprid in *A. glycines*. This study provides a theoretical basis for the further study of the mechanism of action of CYP6s and potential new methods for improving insecticidal efficacy.

## 1. Introduction

The soybean aphid *Aphis glycines* is a major soybean-damaging pest native to Asia and invasive in North America [1,2]. Not only does *A. glycines* damage soybean by performing sucking behavior, but it is also an important vector for transferring many viruses that affect soybean quality and production, such as soybean mosaic virus (SMV), alfalfa mosaic virus (AMV), potato Y virus (PVY) and tobacco ringspot virus (TRSV) [3,4,5,6,7]. Additionally, the growth of black sooty mold fungus on honeydew produced by *A. glycines* affects soybean photosynthesis [8]. Chemical control is the main prevention measure against yield loss inflicted by soybean aphids. However, extensive use of insecticides has led to an increase in production cost, a reduction in natural enemies and resistance of soybean aphid populations.

The invention and development of neonicotinoid insecticides provides valuable, new tools for global agricultural producers to control pests. Imidacloprid is a first-generation neonicotinoid insecticide and is widely used in agriculture [9]. Unlike traditional insecticides, imidacloprid selectively acts on nicotinic acetylcholine receptors (nAChRs) in the insect nervous system by stimulating nAChRs on the post-synaptic membrane, disrupting the normal transmission of the insect central nervous system and paralyzing or even killing insects [10]. In mammals, imidacloprid’s ability to penetrate the blood–brain barrier is poor. Moreover, acetylcholine receptors in the central nervous system and peripheral nervous system have weaker interactions, thus exhibiting low toxicity to mammals [11,12]. Imidacloprid, due to its special mechanism of action, has a great control effect on the piercing–sucking mouthparts of pests such as aphids, planthoppers, leafhoppers and whiteflies. Therefore, it is widely used to control sucking pests affecting crops in China. However, long-term use will lead to increased resistance among *A. glycines* to imidacloprid [13]. It is vital to study the resistance mechanism of imidacloprid in *A. glycines* to increase the insecticidal efficacy of imidacloprid in the field. 

Pest resistance arises from the use of insecticides. Under strong insecticide selection pressure, agricultural pests experience rapid adaptations and evolution [14]. Insect pests develop insecticide resistance through gene mutation, gene amplification, metabolic enzyme activity change, epidermal penetration change, behavioral change and other mechanisms [15,16,17,18]. The enhanced activities of insects’ detoxifying enzymes are ones involving vital mechanisms through which insects develop pesticide resistance. Enhanced detoxification accelerates the detoxification metabolisms of insecticides in insects [19]. P450s are important metabolic systems involved in the detoxification of xenobiotics (pesticides, plant toxins), and almost all types are present in all insect tissues. P450s represent a type of single-chain protein that binds to heme, and it is named after its maximum absorption peak, occurring at a wavelength of 450 nm when combined with reduced CO [20,21,22,23]. P450s make up a gene superfamily composed of multiple gene families mainly divided into four major clades as follows: CYP2, CYP3, CYP4 and mitochondrial P450s. The CYP6 family belongs to the CYP3 clan [22], and many detoxification issues related to P450 genes have been reported to originate from this family, such as *Bemisia tabaci* CYP6CM1 [24], CYP6CM1vQ [25], *Nilaparvata lugens* CYP6AY1, CYP6ER1 [26] and *Mamestra brassicae* CYP6AB56 [27]. 

RNA interference technology selectively inhibits the expression of specific genes in target insects; therefore, it has good application prospects in pest control. Recently, under laboratory conditions, the genetic control of various pests has been achieved by interfering with the expression of key genes in pests [28,29]. However, the effectiveness of RNAi highly depends on the efficiency of nucleic acid molecule delivery to the target. Traditionally, dsRNA has been delivered into organisms that use electroporation, gene gun or microinjection, but these methods may injure model organisms. They are difficult to enact on small insects, and these methods cannot be applied to farmland. The feeding method used to introduce dsRNA is simple to operate, minimizes trauma and has a shorter consumption compared to that of the injection method. However, the efficiency of dsRNA molecules in food used to penetrate the insect intestinal membrane barrier to alter target genes is low. Traditional injection and feeding methods struggle to achieve efficient RNA interference under the conditions of species diversity and genetic diversity. Recently, carrying RNA into the insect body through nanocarriers has provided new ideas for RNA interference experiments. He et al. used acylamide nanocarriers to carry dsRNA into the live Asian corn borer, interfering with the normal expression of chitinase genes and thus delaying development, ceasing molting and even causing the death of the pests [30]. This finding is highly encouraging for the development of RNAi-sourced pesticides.

In this study, 12 CYP6 genes potentially related to imidacloprid resistance were identified through transcriptome sequencing. The mRNA relative expression levels of CYP6s after imidacloprid treatment were used to determine whether they were related to imidacloprid induction. The major CYP6 sequences related to imidacloprid induction were further studied. siRNA was transferred into insects’ bodies to interfere with the expression of the corresponding CYP6 genes in *A. glycines*. The interference efficiency and p450 enzyme activity were detected. Finally, the function of CYP6 in affecting the resistance of *A. glycines* to imidacloprid was verified through a bioassay.

## 2. Materials and Methods

### 2.1. Insect Culture and Insecticide Used

*A. glycines* adults were collected from the Xiangyang Experimental Station of Northeast Agricultural University and cultured in a climate chamber using a potted soybean seedling as the host for many generations without contact with an insecticide. The temperature in the climate chamber was 25 ± 2 °C, the relative humidity was 70% and the photoperiod was 14L:10D. The insecticide used was that of an imidacloprid (Bayer, Leverkusen, Germany) water-dispersible granule, and the content of the effective ingredient was 70%.

### 2.2. CYP6 Sequence Cloning and Analysis

The first to fourth instar nymphs and adult aphids were collected and sequenced to identify the transcriptome (Annoroad, Beijing, China). Twelve *A. glycines* CYP6 sequences were screened from the transcriptome database. Twelve pairs of sequence primers (Appendix A) derived from CYP6 sequences were designed to determine the correctness of the sequences obtained from the database used for PCR amplification. The corrected CYP6 sequences were submitted to the P450 International Nomenclature Committee for Names and registered in GenBank.

### 2.3. CYP6 Expression Pattern Due to Imidacloprid Treatment Induction

*A. glycines* adults were treated with LC50 imidacloprid (37.4 mg/L) for 3, 6, 12, 24 and 48 h. The bioassay method used was defined as follows: We cut fresh soybean leaves to a size of 2 cm × 2 cm and immersed them in the prepared imidacloprid for 10 s. The residual liquid was then removed using filter paper and streaked onto 1% agar medium. We then gently transferred the adults to the leaves, and the plate was inverted and cultured. Each treatment consisted of 3 replicates. Twenty aphids that survived each treatment were collected in one EP tube and snap-frozen in liquid nitrogen and immediately stored at −80 °C until further use. Each treatment was repeated three times.

Total RNA was extracted from *A. glycines* treated with different processing times using Trizol reagent^®^ (Invitrogen, Carlsbad, CA, USA) based on the manufacturer’s instruction. RNA was reverse-transcribed into cDNA using the ReverTra Ace qPCR RT Kit (Toyobo, Shanghai, China). 

A real-time polymerase chain reaction (qRT-PCR) was performed to determine the mRNA levels of CYP6s using THUNDERBIRD SYBR qPCRMix (Toyobo, Shanghai, China) through the CFX Connect Real-Time PCR Detect System (Bio-Rad, Hercules, CA, USA). The cycle conditions were defined as follows: 94 °C for 3 min, followed by 40 cycles of 94 °C for 30 s, 59 °C for 30 s and 72 °C for 30 s. The relative expression levels were calculated through the double-internal-parameter calculation method [31], in which the geometric mean of the CT values of two internal reference genes was combined with the 2^−ΔΔCT^ method [32]. All primer sequences for qRT-PCR are listed in Appendix A. According to Raman Banasl [33], the *A. glycines* samples’ TBP (TATA-box binding protein, JQ654781) and RPS9 (ribosomal protein S9, JQ654782) were used as reference genes. 

### 2.4. Study of the Function of A. glycines CYP6s during Imidacloprid Treatment

Based on the above results, *CYP6CY7* was selected for the detoxification of imidacloprid using RNA interference (RNAi). The siRNAs of the *CYP6CY7* and negative control used in the experiments were designed and synthesized by the GenePharma Company (GenePharma, Shanghai, China), and they are listed in Appendix A. The negative control was a common siRNA that had no RNAi effects on *A. glycines*. siRNAs were dissolved using nucleic acid-free DEPC water, and the final dilution concentration was 20 μM.

In order to convert siRNAs into *A. glycines*, we introduced a transdermal delivery system to carry nucleic acids through nanocarriers and detergents to attach them to the nota of aphids, which smoothly spread around the aphid integument and, finally, penetrated the integument [34]. The SPc was selected as the nanocarrier. SPc and siRNA were mixed at a volume ratio of 1:1 and incubated for 30 min at room temperature. Next, 10% volume of detergent (Liby, Beijing, China) was added to the formulation. Then, 0.1 μL of siRNA/nanocarrier/detergent was applied to the aphid notum for 1, 2, 3, 6, 12, 24 and 48 h. Each treatment was repeated three times. The RNAs of 30 aphids were extracted from each group. The relative expression of mRNA after RNAi was detected through qRT-PCR, and the experiment protocol was the same as that in Section 2.3.

P450 enzyme activities after RNAi were examined using the double-antibody sandwich method based on the insect cytochrome P450 enzyme-linked immunosorbent assay kit (Jiangsu Meibiao Biotechnology Co., Ltd., Yancheng, China). Thirty adult aphids receiving each treatment obtained after RNAi were homogenized on ice in 800 μL of 0.01 M PBS (Beijing Biotopped Technology, Beijing, China) buffer. The homogenate was centrifuged at 4 °C for 10 min at 12,000× *g*. The supernatant was then used to examine P450 activities at a wavelength of 450 nm based on the manufacturer’s instructions. The total protein concentration was determined at 595 nm using a Bradford assay from the TaKaRa Bradford Protein Assay Kit (TaKaRa, Beijing, China). The final unit of the cytochrome P450 enzyme measured was pmol/g.

The sensitivity of *A. glycines* after being subjected to RNAi to imidacloprid was tested using the leaf dipping method, as described above. The adult aphids after 1 h of RNAi were further treated with imidacloprid-impregnated leaves at a concentration of LC50 for 12 h and 24 h, and the mortality was assessed. Twenty aphids were included in each group. Each treatment was repeated three times.

### 2.5. Statistical Analysis

Statistical analyses were performed using SPSS 18.0 software (IBM Corp., Armonk, NY, USA). Differences between groups of means were examined through one-way analysis of variance, followed by Duncan’s multiple range test. A *p* value < 0.05 was considered to be significant.

## 3. Results

### 3.1. Identification of Twelve CYP6 Sequences

Twelve P450 CYP6 sequences of *A. glycines* were identified, nominated and submitted to GenBank. Their molecular characteristics are listed in Table 1. 

The amino acid identity is higher than 40% in the same CYP family [35]. In this experiment, CYP6 amino acids had a sequence identity of 58.97%, indicating that they belonged to the same family. As Figure 1 shows, CYP6CY7 was closer to CYP6CY14 (59.27% identity), followed by CYP6CY16 (55.62% identity) and CYP6CY12 (51.36% identity). CYP6CY48 was closer to CYP6CY20 (65.82% identity), followed by CYP6CY18 (61.91% identity) and CYP6CY8 (56.34% identity).

In general, the insects’ P450 amino acid sequences contained five conserved motifs, namely those of the heme-binding motif (F/YxxGxRxCxG/A), helix-C motif (WxxxR), helix-I motif (AGxxT), helix K motif (ExxR) and “Meander” motif (PxxFxPxxF) [35,36]. As shown in Figure 1, 11 of the 12 CYP6 amino acid sequences in the *A. glycines* contained all the five conserved motifs, with the exception being CYP6CY12, while CYP6CY12 contained four conserved motifs, with the exception being the helix-I motif (AGxxT).

### 3.2. Differential Expression of CYP6s in Response to Imidacloprid Treatment Induction

To identify the specific CYP6s that contribute to imidacloprid induction in *A. glycines*, the relative expression levels of 12 CYP6s from these aphids were evaluated using qRT-PCR. The results showed that the expressions of 12 CYP6s significantly increased for at least one time point within 48 h after the induction of imidacloprid at an LC50 concentration. The genes *CYP6CY7*, *CYP6CY8*, *CYP6CY9*, *CYP6CY14* and *CYP6CZ1* responded quickly at 3 h, while the genes *CYP6CY12*, *CYP6CY16*, *CYP6CY18*, *CYP6CY20*, *CYP6CY48*, *CYP6DB1* and *CYP6DD1* exhibited a relatively slower response. Among them, the relative expression levels of *CYP6CY7* have the highest upregulation, which is more than 3-fold compared to those observed for the non-induction of treatment. The relative expression level of *CYP6CY7* was highest at 12 h after induction, which was 3.55-fold higher than that of non-induction, followed by *CYP6CY48*, which was 3.36-fold higher than that of non-induction at 48 h after induction (Figure 2). Thus, *CYP6CY7* might be the main gene that provides resistance to imidacloprid stress. 

### 3.3. Relative Expressions of CYP6CY7 after RNAi

The inhibition efficiencies of *CYP6CY7* were determined using qRT-PCR after RNAi. The results demonstrated that the relative expression levels of *CYP6CY7* decreased by 58.49%, 73.80% and 69.30% at 3 h, 6 h and 12 h after treatment (Figure 3), which indicated that mRNA expression level of *CYP6CY7* was inhibited at different time points after treatment. 

### 3.4. P450 Enzyme Activities after RNA Inhibition

The enzyme activities of P450 were examined, as shown in Figure 4; after the inhibition of *CYP6CY7*, the enzyme activity of P450 was significantly decreased at 6 h, and it was decreased by 19.38% and 21.96% at 12 h and 24 h. The RNA inferences took effect after 6 h, and the effects were relieved after 48 h.

### 3.5. Sensitivity of A. glycines to Imidacloprid Treatment after RNA Inhibition

We measured the sensitivity of *A. glycines* to imidacloprid treatment after *CYP6CY7* inhibition through leaf dipping methods. When *A. glycines* were exposed to imidacloprid-containing leaves for 12 h, the mortality of *A. glycines* inhibited with siRNA of *CYP6CY7* increased by 13.13%. After exposure to imidacloprid-containing leaves for 24 h, the proportion of aphids inhibited with siRNA of *CYP6CY7* also significantly increased (Figure 5).

## 4. Discussion

Regarding insect detoxification, metabolic systems are the key mechanisms for developing insecticide resistance [37]. Insect cytochrome P450 enzymes are present in organisms and are widely implicated in the detoxification of xenobiotics. The soybean aphid is a global agricultural pest that causes huge economic losses, and its increasing resistance to pesticides in recent years has made the effective control of this pest a problem that urgently needs to be solved. In this study, 12 cDNA sequences from the CYP6 family of the soybean aphid were obtained from the transcriptome database, named by the International P450 Nomenclature Committee and submitted to the NCBI database. The 12 amino acid sequences contain conserved regions common to insect P450s, thus being consistent with the characteristic sequences of this family. Their sequence identity was 58.97%, indicating that the subfamily sequence in the P450 family was relatively conserved.

The cytochrome P450 is inducible after exposing insects to a certain degree of heterologous stress, and a quick and significant increase in the levels of cytochrome P450 genes results in enhanced detoxification [38]. 

Multiple families and subfamilies of insect P450 genes exist, of which the CYP3 and CYP4 families are considered to be most closely related ones to drug resistance [39,40]. Indeed, the *CYP4G8* gene of *Helicoverpa armigera* was overexpressed twice as much in the pyrethroid insecticide-resistant strain [41], and five P450 genes of the CYP4 family that may be associated with deltamethrin resistance were also previously identified in *Culex pipiens* [42]. As research in this field progressed, the functions of the CYP6 genes were discovered, with the genes *CYP6CY7* and *CYP6CY21* being related to α-cypermethrin cross-resistance in *Aphis gossypii* Glover [43], while the knockdown of one CYP4 gene and five CYP6 genes dramatically increased the sensitivity of *A. gossypii* to thiamethoxam or imidacloprid treatment [44]. The CYP6 family may be induced at different concentrations and different pesticide treatment times; in this study, the 12 CYP6 genes exhibited different upregulated expression patterns after imidacloprid treatment at different processing times. 

Upon comparing multiple experiments at home and abroad, the relative expression levels of *CYP6AB56* and *CYP4M51* in *Mamestra brassicae* induced with deltamethrin at LD50 for 3 h, 6 h, 12 h, 24 h and 48 h showed a trend of an initial increase (reaching a maximum at 12 h) before gradually decreasing [27]. In imidacloprid-resistant strains of B and Q biotypes of *Bemisia tabaci*, the expression levels of *CYP6CM1* showed a certain temporal effect [24], and this phenomenon was also observed in the *CYP6AY1* and *CYP6ER1* from the brown planthopper [26,45]. With the overexpression of CYP6 genes, induction caused activity enhancement in the detoxification system, increasing metabolic defenses. 

RNAi technology is a universal method employed to identify resistance genes, and Bautista reported that after specific interference with the *CYP6BG1* gene in the diamondback moth, the P450 enzyme activity decreased by 25.5%, thus reducing the larvae’s resistance to cypermethrin [46]. RNAi knockdown of *CYP6BG1* in diamondback moth larvae resulted in a 45.5% decrease in P450 enzyme activity, thus increasing sensitivity to chlorantraniliprole by 26.4% [47]. The results showed that using RNAi with *CYP6CY7* significantly decreased P450 activity by up to 21.96%. A further bioassay showed that the sensitivity of soybean aphids to imidacloprid significantly increased after the interference. This result indicated that the metabolic activity of the proteins encoded by the *CYP6CY7* gene plays an important role in P450 metabolic activity, while the metabolic activity plays an important role in this insect’s insecticide detoxification mechanism. 

After the RNAi of *CYP6CY7*, the interference effect on the soybean aphid did not appear until 3 h, and it remained at 12 h. The activity of cytochrome P450 significantly decreased after 12 h and lasted until 24 h, and the sensitivity of aphids to imidacloprid increased significantly after 24 h. The efficiency of RNAi depends on various factors, such as the mechanisms used to inhibit or interfere with RNAi in cells as well as degradation by nucleases or immune responses used to recognize and eliminate foreign dsRNA or siRNA molecules, and some cells may also have low expression or activity of RNA-induced silencing complex (RISC) components or other factors crucial for RNAi [48]. Due to differences in the onset times of RNAi, the changes in the corresponding enzyme also exhibit time specificity. At the same time, the sensitivity of soybean aphids to imidacloprid treatment caused by the detoxification function and mediated by different genes also exhibits time specificity. However, further research is needed to understand how the expression of genes mediates the time-specific changes in enzymes used for field application guidance.

RNAi technology can selectively interfere with target genes in target insects, effectively alleviating long-term concerns about pesticide residues and food safety; therefore, it shows high potential in pest control. The optimal effect of RNAi-mediated gene silencing is difficult to achieve, especially in aphids. Due to the fact that the abdominal cavity is filled with bodily fluids, the direct injection of double-stranded RNA is not only complicated but can also lead to the death of experimental insects due to mechanical injury, post-injection infection and other issues, thereby resulting in inaccurate test results. 

It is difficult to eliminate malnutrition-related errors by feeding soybean aphids artificial diets for dsRNA interference. Additionally, the efficiency of the dsRNA molecules penetrating the membrane barrier of insect intestine to reach the target genes is reduced. Directly soaking soybean aphids in a dsRNA aqueous solution can reduce mechanical damage and achieve effective interference, but soybean aphids often escape from the soaking solution, and the soaking time required to achieve effective delivery cannot be guaranteed [34]. In addition, complete immersion in a high amount of dsRNA solution causes significant economic loss. Thus, we chose to use the body wall permeation method by introducing the nano-carrier into the insect body to not only ensure the survival rate of the experimental soybean aphid but also to create the potential for practical production.

dsRNA degrades its mRNA homologous, a process known as RNA interference. dsRNA was cleaved into short interfering RNAs (siRNAs) and hybridized with homologous mRNA, which induced its degradation. In general, both dsRNA and its metabolite, siRNA, can induce RNAi effects in different species. Compared with dsRNAs, siRNAs are easy to synthesize and to mass-produce at a low cost. siRNAs can also be designed to prevent their degradation and preserve stability [49,50]. siRNAs may also be developed as a specific nucleic acid insecticide [51]. Researchers have previously reported many successful RNAi experiments that used siRNA [52,53]. This experiment introduced siRNAs into soybean aphids and suppressed the relative expression level of the *CYP6CY7*, as well as the enzyme activities of P450s, thereby improving the sensitivity of soybean aphids to imidacloprid. Therefore, it provides a theoretical basis for RNAi experiments and the development of new insecticides.

## 5. Conclusions

In this study, we identified and characterized 12 P450 CYP6 genes from *A. glycines*. The overexpression of these genes after the induction of imidacloprid treatment at an LC50 concentration provided new information about the role of CYP6s in determining resistance to insecticides in *A. glycines*. Through siRNA interference by nanocarriers, we interfered with the expression of *CYP6CY7*, highly increasing the susceptibility of aphids to imidacloprid; the P450 enzyme activity was inhibited by 21.96%, the insecticidal efficacy of imidacloprid treatment on *A. glycines* increased and the mortality of the interfered *A. glycines* in response to imidacloprid treatment increased by 14.71%. Compared to dsRNA, siRNAs are easy to synthesize and to mass-produce and are decreasing in cost. Small-molecule siRNAs can also be designed to prevent degradation and improve stability, and they also have the potential for development as a specific nucleic acid insecticide. Therefore, this study provides a theoretical basis for further research into the potential metabolic detoxification mechanisms of CYP6s and the development of new synergists to increase insecticidal efficacy. 

## Figures and Tables

**Figure 1 insects-15-00188-f001:**
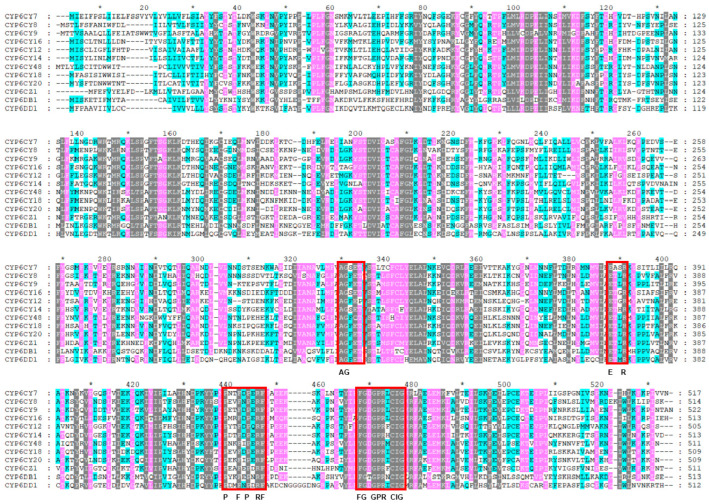
The alignment of the amino acid sequences of 12 CYP6 genes from *A. glycines*. The conserved motifs are shown in red squares. The black area indicates an identity of 100%, the pink area indicates an identity of 75% or more and the blue area indicates an identity of 50% or more. * indicates the position of the number next to it ±10.

**Figure 2 insects-15-00188-f002:**
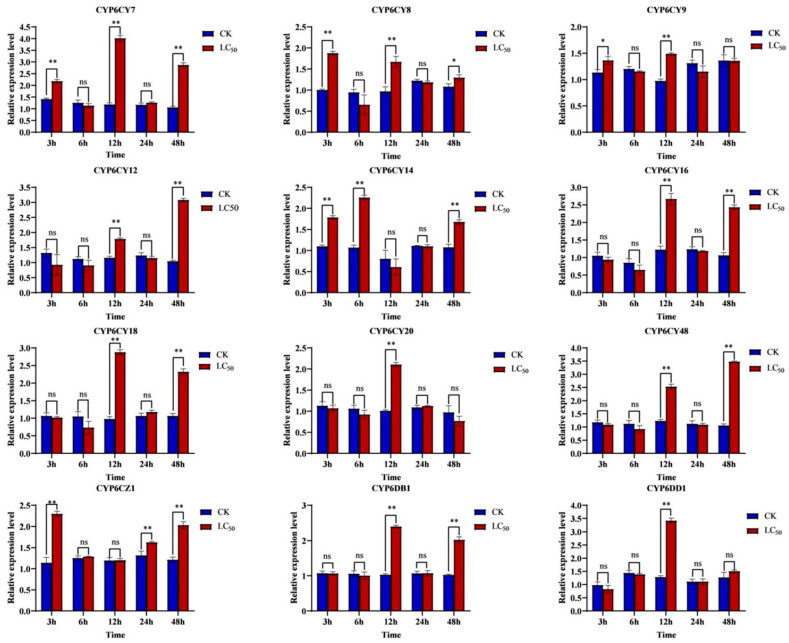
Effects of imidacloprid induction on the expression of CYP6s by *A. glycines*. The relative expression levels of CYPs were examined after imidacloprid induction at 3, 6, 12, 24 and 48 h. *TBP* and *RPS9* were used as the internal reference genes. Expression levels of CYPs were analyzed using the 2^−ΔΔCT^ method and by calculating the means ± SEs. Statistical differences at *p* < 0.05 are indicated by *, *p* < 0.01 are indicated by ** and the ns represents no significant differences.

**Figure 3 insects-15-00188-f003:**
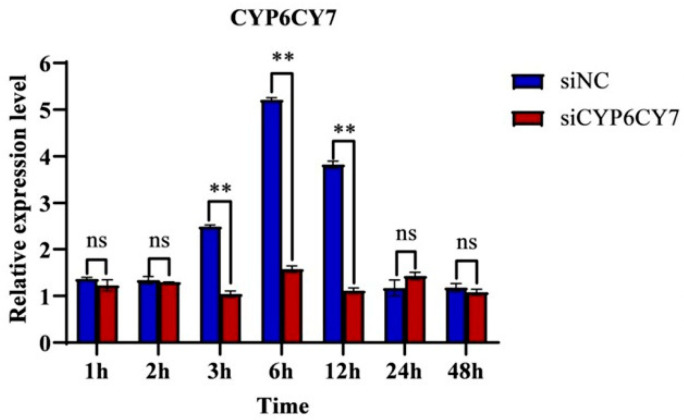
Expression levels of *CYP6CY7* after the siRNA treatment of *A. glycines* at different times. The relative expression levels of the gene were compared with the NC after RNA interfered for 1, 2, 3, 6, 12, 24 and 48 h. The negative control was a kind of siRNA that showed no RNAi effects in *A. glycines*. The results are the mean ± SE of three independent replications. ** represents significant differences between different treatments (*p* < 0.01) and the ns represents no significant differences.

**Figure 4 insects-15-00188-f004:**
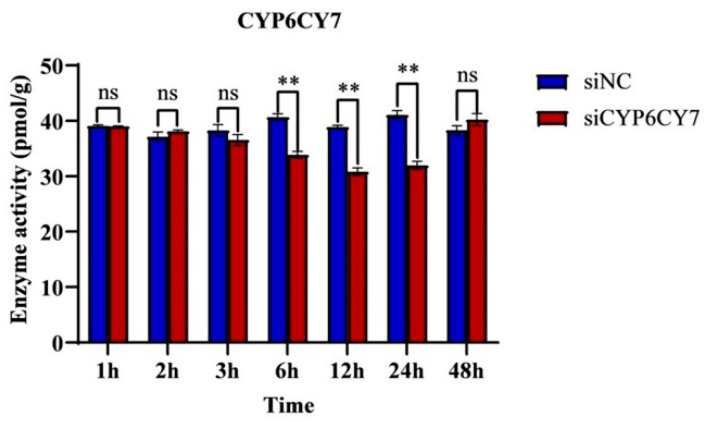
Enzyme activities of P450 in *A. glycines* after *CYP6CY7* inhibition. RNA interferes with *CYP6CY7* and NC for 1, 2, 3, 6, 12, 24 and 48 h. The results are the mean ± SE of three independent replications. ** represents significant differences between different treatments (*p* < 0.01) and the ns represents no significant differences.

**Figure 5 insects-15-00188-f005:**
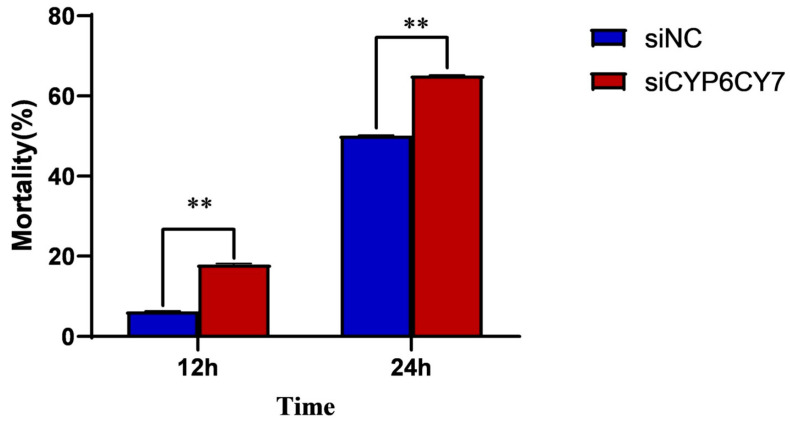
Mortality of *A. glycines* treated with imidacloprid LC_50_ after *CYP6CY7* inhibition. The results are the mean ± SE of five independent replications. ** represents significant differences between different treatments (*p* < 0.01).

**Table 1 insects-15-00188-t001:** The characteristics of CYP6s identified from *A. glycines*.

Gene Name	Accession No.	Length (bp)	Length (aa)	pI	Mw (kDa)
*CYP6CY7*	MN055997	1735	517	8.40	59.20
*CYP6CY8*	MN056000	1913	514	8.60	59.50
*CYP6CY9*	MN055999	2071	522	8.34	59.44
*CYP6CY12*	MN055998	1686	505	8.68	58.21
*CYP6CY14*	MN056003	2468	513	7.56	58.96
*CYP6CY16*	MN055995	1769	513	8.38	59.27
*CYP6CY18*	MN056002	1556	508	8.76	59.38
*CYP6CY20*	MG829856	2040	509	9.10	59.20
*CYP6CY48*	MN055996	1639	511	8.87	59.48
*CYP6CZ1*	MN055993	1837	511	8.86	59.37
*CYP6DB1*	MN056001	2287	513	9.11	59.58
*CYP6DD1*	MN055994	2108	512	8.27	58.10

## Data Availability

The data presented in this study are available upon reasonable request from the corresponding author.

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
