# Peer review of "RNA Interference Reveals the Impacts of CYP6CY7 on Imidacloprid Resistance in Aphis glycines"

_insects, 2024, doi:10.3390/insects15030188_

Round 1

Reviewer 1 Report

Comments and Suggestions for Authors

Comments to the manuscript: RNA Interference Reveals the Impact of CYP6CY7 and 2 CYP6CY48 on Imidacloprid Resistance in Aphis glycines

Soybean aphids are the major pests of soybean crops. In a study conducted by Li et al., the focus was on P450s genes, their expression, RNAi targeting two main P450s genes, and evaluating the impact of imidacloprid following subdermal exposure of aphids to siRNA. The methods employed in this study appeared to be appropriate. However, some comments required further explanation. The suppression of mRNA and increased toxicity observed after administering siCYP6CY7 and siCYP6CY48 suggest a role for these two P450s genes in aphid resistance to neonicotinoid insecticides. Additionally, increased suppression of mRNA causes mortality under insecticide treatment. P450s play a crucial role in aphid resistance to neonicotinoids. However, it remains unclear whether these colonies are resistant to neonicotinoids. This was indicated by the low mortality rate observed (potentially 15% after 24 h of treatment). While the authors highlighted the role of P450s genes in neonicotinoid resistance, the higher expressed gene was responsible for the low mortality rate. This study has some limitations. Li et al should provide more information on the resistance levels of this aphid colony. It is also recommended that a professional editor reviews the English language used in the manuscript.

Please see below other comments: 

L83-use italics for scientific names

L85 rephrase.

L99 year of the citation

L129 Use only one decimal.

L147 refrence

L151 Rephrase a ..kind of commercial..

L157 Define SPc

L223 Explain the reason of using the LC50 instead of another LC

L255 correct increased

L257 Figure 5 the only significate mortality after 24 hours was in the siCYP6CY7 treatment. The impact of administration of this siCYP6CY7 seems to be very low-about 15% at 24 h. 

L305-307..in the paragraph…. This indicated that the metabolic activity of the proteins encoded by the CYP6CY7 and CYP6CY48 genes play an important role in P450 metabolic activity, and their metabolic activity plays an important role in the insecticide resistance mechanism of this insect. However, there is no evidence that this aphid colony is resistant to imidacloprid. Please address the weaknesses of this study.     

L309 After.. what?

L323 This sentence is unclear: Aphid is an important pest, but a typical material of gene interference instability.

L330 Correct…and intestinal cell membrane to target the target genes is reduced.

L333 In addition, complete immersion in a large amount of dsRNA solution results in significant loss. Loss of what?

Ln 9 – make sure to write scientific names correctly; same comment for Ln 280, 291

Ln 22 – change “make clear” to elucidate

Ln 49 – 52 Rewrite this sentence to be less wordy and lengthy. Suggestion: Chemical control is the main prevention measure against yield loss inflicted by soybean aphids. However, extensive use of insecticides has led to an increase of production cost, reduce of natural enemies, and resistance of soybean aphid populations.

Ln 53 – remove word “commercial”

Ln 60 – use period rather than comma before Moreover.

Ln 85 – 86 – rewrite this sentence to be more concise.

Ln 90 – 92 – change from “Traditionally, people have introduced dsRNA through physical methods such as electroporation, gene gun and microinjection, but the scope is limited and it is prone to damage to organisms” to “Traditionally, dsRNA have been delivered into organisms using electroporation, gene gun and microinjection, but these methods may injure model organisms”

Ln 108 – change transfer to transferred.

Ln 188 – was this technical grade or a brand of imidacloprid?

Method

Ln 129 – How were aphids treated? Potter sprayer? Treated plants and aphid allowed to feed? Please elucidate in detail

Ln 147 – placed reference for CYP6CYT7 and CYP6CY48

Results

Ln 259 – where are the lower case in the images?

CYP6DD1 also had relative expression ~3.5 why did not choose this also?

Discussion

Explain more on different genes also exhibits time specificity on Ln 317.

What about other P450 genes related to neonicotinoids resistances? Is it only from one family or maybe other closely related families? Does it change within certain insect families or taxa?

Comments on the Quality of English Language

Reviewer 2 Report

Comments and Suggestions for Authors

Li et al investigated a consequence of CYP6CY7 and CYP6CY48 knockdown on imidacloprid resistance in Aphis glycines. Knockdown of CYP6CY7 and CYP6CY48 improved mortality against imidacloprid in Aphis glycines. This research provided valuable information that CYP6CY7 and CYP6CY48 somewhat play a role in the mode of action of imidacloprid resistance. However, the experimental design of this study needs to be improved.

Even though the mortality was increased with the knockdown of CYP6CY7 and CYP6CY48, it does not show the drastic changes in morality. Authors need to think about What would be the factors for that?

I believed that other CYP6s or CYPs would have more functions in imidacloprid resistance.

Line 121: need a justification for choosing CYP6CY7 and CYP6CY48 other than "increased more than 3-fold". This fold change comparison is not an acceptable way of choosing a target gene.

Or

Knockdown efficiency may not reach the ideal point to see the final net-effect, mortality by imidacloprid.   

The author needs to check the RNAi efficiency whether siRNA is sufficiently knockdown the target gene.

Or

The author might need to try all CYP6 genes to find the best effector.  

 Here are some comments: 

Line 148: need to explain about previous research. reference? 

 Line 187 : Table 1  -->  What is the point of having "pI" and "Mw" are they important? I would rather put the potential functions of this gene. 

 Line 194- 198 :   those motif were not properly indicated in the Figure 1.

Reviewer 3 Report

Comments and Suggestions for Authors

The theme of the Manuscript fits in the scope of the Journal. The manuscript describes some important works. The study not only reveals the potential metabolic detoxification mechanism of CYP6CY7 and CYP6CY48 in Aphis glycines, but also provides a theoretical basis for the creation of new insecticide synergists. The manuscript needs minor revision before acceptance can be considered.

 The following are some suggestion.

Line 133 & 137:  There are problem about sample collection. Are the samples from the whole body of the adult aphids or the fifth instar aphid? please clarify.

Line 142:  what do you mean by “the different treatment” ?

Line 152:  “All primer sequences for qRT-PCR are listed in Table 1.” you may mean Table S1 in this sentence.

Line 158:  In this study, to avoid the off-target effect, it's better to use two separate siRNA to confirm its effect in the cells.

Line 172:  the same as that in 1.3.

Line 175:  Check for the English name of the company.

Line 176:  "μl", use μL.

Line 299:  there had a write mistake. “midacloprid”

Line 334:  references missing in discussion. Please add in appropriate positions.

Comments on the Quality of English Language

The English is fair and in most places does not obscure the authors' intended meaning. I suggest that the author carefully revise the Manuscript to make it clearer and more comprehensive.

Round 2

Reviewer 2 Report

Comments and Suggestions for Authors

The reference for selecting CPY6 seems to have been added. However, it is still vague as to why selecting two genes out of 12 is not appropriate, and that is why the results are not showing much difference compared to other studies. Even if the expression level of a gene has increased threefold, it may not directly impact resistance. Even if the gene expression has increased twofold; it could play a more important role than a threefold increase. Do you have any plans to try the remaining 10 as a screening concept? I am sure that other genes will show better effects on resistance. 

The story flows better when RNA seq data is provided at the beginning as basic data to choose the candidate genes. Then, qPCR to confirm the RNA seq data and check the timely expression. Then, knockdown of some upregulated genes to check the involvement of resistance.  
